# Is Nutritional Ultrasound as Useful and Accurate as Computed Tomography to Assess Sarcopenia in Cancer Patients? A Systematic Review

**DOI:** 10.3390/cancers17223683

**Published:** 2025-11-18

**Authors:** Luis M. Luengo-Pérez, Claudia García-Lobato, Lucía Lázaro-Martín, Juan D. Gallardo-Sánchez, Marta M. Guijarro-Chacón

**Affiliations:** 1Medical Sciences Department, Faculty of Medicine and Health Sciences, University of Extremadura, 06006 Badajoz, Spain; juandiego.gallardo@salud-juntaex.es; 2Clinical Nutrition and Dietetics Unit, Endocrinology and Nutrition Section, Badajoz University Hospital, 06008 Badajoz, Spain; claudia.garcial@salud-juntaex.es (C.G.-L.); martamaria.guijarro@salud-juntaex.es (M.M.G.-C.); 3Endocrinology and Nutrition Service, Cabueñes University Hospital, 33394 Gijón, Spain; lucia.lazaro@sespa.es; 4Internal Medicine Service, Badajoz University Hospital, 06008 Badajoz, Spain

**Keywords:** sarcopenia, neoplasms, ultrasonography, tomography, X-Ray Computed

## Abstract

Sarcopenia is an important issue in patients with cancer, as it worsens response to cancer treatment and prognosis. Low muscle mass is a criterion for sarcopenia that must be evaluated in patients at risk of malnutrition, including those with cancer. To date, muscle mass is mainly assessed by computed tomography (CT) in an opportunistic way, as it is performed to evaluate cancer stage and response to treatment. Nutritional ultrasound (NU) is an emergent technique to assess muscle mass, and the objective of this systematic review is to evaluate its usefulness and accuracy for this purpose, comparing it with CT. A total of 1011 patients with cancer were included in the six articles found in this review, which shows that NU is useful and accurate enough to be used in the clinical setting to evaluate low muscle mass in the follow-up of patients with cancer.

## 1. Introduction

Malnutrition is a common and clinically significant complication in cancer patients, with a high prevalence that varies depending on tumor type, disease stage, and treatment. Its diagnosis relies on clinical and functional tools that help to identify alterations in dietary intake, body weight, body composition, and inflammatory status [1,2]. In the oncological context, malnutrition is often associated with a dysregulated metabolic response, characterized by increased catabolism, and reduced protein synthesis [3]. Among the most frequent causes are anorexia induced by the tumor or its treatment, gastrointestinal side effects, systemic inflammation, and cancer cachexia [4]. These factors contribute to a progressive loss of muscle mass, even in patients with apparently normal body weight. The clinical consequences are significant: increased treatment toxicity, higher risk of postoperative complications, reduced quality of life, and decreased survival [5]. Early nutritional intervention, particularly through formulas tailored to the patient’s metabolic profile, has been shown to improve clinical and functional parameters, reduce complications, and support recovery [6]. Therefore, nutritional assessment should be an integral part of cancer care from the outset.

Sarcopenia, defined as the progressive loss of muscle mass, strength, and function, is especially relevant in cancer patients due to its impact on prognosis and treatment tolerance [6]. Although initially associated with aging, it is now recognized as a multifactorial condition that can occur at any stage of life in the presence of chronic disease [7]. Diagnosis is based on international criteria that combine the evaluation of muscle strength, muscle quantity, and physical performance [8]. Low muscle mass is defined with cutoff points for dual X-ray absorptiometry (DXA), bioelectrical impedance analysis (BIA), and computed tomography (CT) [2], while nutritional ultrasound (NU) was considered by the European Society for Clinical Nutrition and Metabolism (ESPEN) as useful for patient follow-up for repeated measurements, but lacking standardization and validation studies [9].

In oncology, sarcopenia may coexist with malnutrition or appear independently, even in patients with normal or excess body weight. Chronic inflammation, physical inactivity, treatment toxicity, and altered protein metabolism are key contributors to its development [10]. Clinically, sarcopenia is associated with increased risk of surgical complications, reduced chemotherapy tolerance, higher mortality, functional decline, and loss of autonomy [11]. Early detection allows for the implementation of nutritional and functional strategies that may improve clinical outcomes and quality of life.

Computed tomography has become the reference method for assessing muscle mass in cancer patients due to its high precision and availability in clinical settings [12]. Cross-sectional analysis, particularly at the level of the third lumbar vertebra, enables estimation of total muscle mass using specialized software, with high reproducibility and diagnostic sensitivity. This technique offers advantages such as retrospective analysis of images already obtained for oncological purposes, avoiding additional procedures. However, it also presents limitations: it requires specific training, access to segmentation tools, and involves radiation exposure, which limits its use for repeated nutritional monitoring [13]. In recent years, NU has emerged as a more accessible, safe, and dynamic alternative for evaluating muscle mass. Although CT remains the gold standard, NU has shown comparable results in certain clinical contexts, opening new possibilities for its integration into routine practice [13,14].

Nutritional ultrasound is used as a non-invasive tool for assessing sarcopenia, allowing to measure both muscle quantity and quality. The most studied parameters include rectus femoris muscle thickness (RFT), cross-sectional area (RF-CSA), and echogenicity. These parameters exhibit a moderate to high correlation with reference methods, such as DXA and BIA, and demonstrate good inter- and intra-observer reliability [15,16].

The RFT and RF-CSA, measured by ultrasound, have demonstrated moderate to high diagnostic ability in identifying low muscle mass/sarcopenia, with the area under the curve (AUC) values ranging from 0.76 to 0.90, according to various studies [15,17]. In addition, echogenicity, which reflects muscle quality and fat infiltration, can complement quantitative assessment, although its diagnostic accuracy is lower [17,18]. This technique is mainly performed on muscles such as the rectus femoris and gastrocnemius, as they are anatomically accessible, have an easily identifiable architecture, and are representative of appendicular muscle mass [16,19].

Muscle ultrasound is a growing tool for diagnosing sarcopenia in cancer patients, as it allows for rapid, non-invasive bedside evaluation. It has also been used to predict relevant clinical outcomes, such as postoperative complications, hospital stay, and survival [20,21].

The advantages of muscle ultrasound for assessing sarcopenia in cancer patients include its low cost, portability, absence of radiation, and ease of use at the bedside, making it particularly useful in critically ill patients or those with limited mobility. However, it has limitations, including the diverse standardization of protocols, a lack of universally validated cutoff points, and lower diagnostic accuracy compared to CT, especially in the assessment of muscle fat infiltration [22]. The lack of universally validated cutoff points for low muscle mass is being solved, as cutoff points for hospitalized patients at risk of malnutrition [19] and for oncology patients [23] have now been defined.

On the other hand, CT is considered the gold standard for the assessment of sarcopenia in oncology, as it allows for highly accurate quantification of muscle mass and assessment of muscle quality through density and intramuscular fat. In addition, most patients diagnosed with cancer undergo CT for disease staging, and the images can be used to assess sarcopenia without the need for additional tests. Its disadvantages include, as mentioned previously, high cost, exposure to ionizing radiation, and the need for specialized software and time for muscle segmentation at L3, as well as the absence of universal standardized cutoff points for sarcopenia in some contexts [24,25].

The aim of this systematic review is to evaluate whether NU offers comparable accuracy to CT in assessing sarcopenia in cancer patients, and whether it could serve as a useful clinical tool given its wider availability. The main objective of the review is to assess diagnostic accuracy of NU vs. CT, expressed as AUC with 95% confidence intervals, when available, and the secondary objectives are the correlation between NU and CT and the prevalence of low muscle mass using prespecified or estimated cut-offs.

## 2. Materials and Methods

### 2.1. Protocol and Registration

This systematic review was developed following the checklist from “preferred reporting items for systematic reviews and meta-analyses. 2020” (PRISMA 2020) [26,27]. The protocol was registered at Open Science Framework (OSF), with identification 8r3vs, where PRISMA 2020 checklist is available (https://osf.io/8r3vs/ (accesed on 8 November 2025)).

### 2.2. Selection Criteria

The inclusion and exclusion criteria for selecting articles were developed according to the PICO question: Patients (with cancer), Intervention (nutritional ultrasonography, NU), Control (computed tomography, CT), and Outcome (sarcopenia prevalence, correlation between NU and CT).

Thus, the inclusion criteria were patients with neoplastic diseases and the evaluation of sarcopenia with both NU and CT.

The exclusion criteria were studies involving non-cancer diagnoses, lack of muscle mass assessment, or absence of NU and/or CT evaluation.

### 2.3. Search Strategy

A literature search was conducted in PubMed and Scopus databases on 18 May 2025 and repeated on 20 August 2025.

In PubMed, medical subject headings (MeSH) “Sarcopenia”, “Neoplasms”, “Ultrasonography”, and “Tomography, X-Ray Computed” were employed with the Boolean operator AND in all cases. Full search equation in PubMed was ((“Sarcopenia” [Mesh]) AND “Neoplasms” [Mesh]) AND ((“Ultrasonography” [Mesh]) AND (“Tomography, X-Ray Computed” [Mesh])). No limits in date range nor filters or language restrictions were employed in the search. In the Scopus database, the same words were employed for searching articles, which included all of them (Boolean operator AND) in the title, abstract, or keywords. The full search equation is (TITLE-ABS-KEY(“Sarcopenia”) AND TITLE-ABS-KEY(“Neoplasms”) AND TITLE-ABS-KEY(“Ultrasonography”) AND TITLE-ABS-KEY(“Computed Tomography”)).

In order to avoid missing other relevant studies, a secondary search strategy was developed by screening the references included in the previously selected articles and on Google Scholar, where an additional article was found [28]. The last article, by López-Gómez et al. [23], was found from alerts of scientific journals when it was published at the end of September 2025. The search process is summarized in Figure 1.

### 2.4. Study Selection and Data Collection

Every article identified in the search process was included in the screening process, after duplicated articles were removed. Two reviewers (L.M.L.-P. and M.M.G.-C.) independently screened selected article titles and abstracts for their relevance. Automation tools were not used in the process. Full-text articles from relevant studies were downloaded and then assessed if they were eligible according to the criteria in Section 2.2.

Data collection from every included article was performed by two authors (L.M.L.-P. and M.M.G.-C.) independently in a standard data extraction form, who then compared the two forms and included all collected data in a single definitive form, after resolving the disagreements between both reviewers. All forms included first author, country, year, study design, method, sample characteristics, sarcopenia or low muscle mass prevalence (outcome), and correlation between NU and CT (outcome). Data collected were the ones present in the published articles, without including or calculating missing data, and are summarized in Table 1. Kappa was included when reported by authors.

### 2.5. Assessment of Risk of Bias in Every Selected Study

Two authors (L.M.L.-P. and M.M.G.-C.) independently evaluated the risk of bias of individual selected studies according to the National Institutes of Health (NIH) Quality Assessment Tool for Observational Cohort and Cross-Sectional Studies [29], including the following items: (1) statement of research question or objective, (2) definition of study population, (3) rate of eligible persons, (4) inclusion and exclusion criteria, (5) sample size justification, (6) outcome measures definition and implementation, (7) blinding of outcome assessors, and (8) adjustment of confounding variables. The rest of the items were not included, as they were not applicable (e.g., level of exposure) or not discriminatory (outcomes measures implemented consistently). Whenever there was a disagreement in the assessment of risk of bias between both reviewers, it was solved by consensus.

## 3. Results

### 3.1. Study Selection

A total of eight articles were identified in Scopus and five in the PubMed search. Three additional articles were found in the secondary search process. Figure 1 shows the flow diagram of the search process, following the PRISMA 2020 model [26].

Titles and abstracts of the 12 articles selected were screened for relevance [8,13,22,23,28,30,31,32,33,34,35,36], after revoving 4 of them because they were duplicated [8,13,22,31].

Six of the twelve articles were excluded: two because the patients included did not have cancer [30,32], and in four studies, the patients were not assessed by nutritional ultrasound [33,34,35,36].

A total of six full-text articles were then assessed for eligibility [8,13,22,23,28,31], and none of them were excluded because all fulfilled the inclusion criteria, and they did not have exclusion criteria.

### 3.2. Study Characteristics

The six studies evaluated in the review included 1011 patients with cancer, 579 (57.27%) male patients. Their main characteristics are shown in Table 1.

**Table 1 cancers-17-03683-t001:** Summary of the studies included in the systematic review.

Reference (Country)	Study Design	Method	Sample Size (F/M, Age, BMI, Stage/Type)	Sarcopenia (Low Muscle Mass)	Correlation US vs. CT	Risk of Bias ^1^
Sousa IM et al., 2025 (Brazil) [22]	Secondary cross-sectional analysis of cohort studies with prospective data collection	CT-CSA vs. US BMT &/or TMT	120 hospitalized patients with cancer (53.3% female, age 62 (55–70), 40.9% overweight, 65.8% digestive, 85.8% stages III-IV)	CT: low CSA 49.2% US: low BMT 30.8%, low TMT 18.3%, low BMT + TMT in 48.3% (34% sarcopenia) Female, all lower predictive accuracy (AUC = 0.50). Male, TMT highest accuracy (AUC = 0.78), combined BMT + TMT (AUC = 0.76)	TMT and BMT + TMT vs. CSA (R^2^ = 0.35) *; +accuracy (AUC > 0.70); moderate agreement w. CSA (k = 0.48)	Moderate
Jiménez-Sánchez A et al., 2024 (Spain) [8]	Cross-sectional	CT-SMA at L3 vs. RF-CSA/RF-MT/QMT	156 consecutive colorectal outputs (51.9% male, age 65.2 (SD 13.6), overweight 62.9%, IIIB 20.8%)	CT: 1 vs. 4/156 (Van Vogt vs. Dolan)/US: 2 vs. 0/156 (RF-CSA vs. RF-MT)//Muscle atrophy CT: 10 vs. 76/156/US: 13 vs. 16/156, respect.	Muscle atrophy. CT (Van Vogt) vs. US-MT/CSA k = 0.165 */k = 0.109 (ns)	Moderate
de Lellis J et al., 2025 (Brazil) [13]	Prospective cohort study	QMT (2/3, VALIDUM method, +/− compression) vs. L3 CT-SMMI	88 patients from oncological intensive unit (male 54.5%, age 60.6 (SD 13.0), BMI 25.1 (SD 6.3), 47.7% digestive, TNM NR)	CT (Toledo threshold): 63.6% vs. US. QMT AUC 0.706 (*p* < 0.001)/0.667 (*p* = 0.005) (2/3 +/− compression) Similar at 1/2	CT vs. US-QMT (2/3 compression) ≤1.29 cm: PPV 85.4%, NPV 62.5%, agreement 75% QMT ≤ 1.29 cm more likely lower CT-SMMI (OR 0.96, 95% CI: 0.92–0.99) *	Low
Guirado-Peláez P et al., 2024 (Spain) [31]	Retrospective cross-sectional study	CT-SMI (L3) vs. US-RF-CSA	267 colorectal cancer patients (male 61.8%, age 68.2 (SD 10.9), BMI 26.8 (SD 4.93), III-IV 39.7%)	Low SMI: Martin’s criteria, 43.8%/Prado’s criteria, 49.8% low SMI”	CT-SMI (L3) vs. US-RF-CSA: r = 0.56 (*p* < 0.001)	Moderate
González-Bollos M et al., 2024 (Spain) [28]	Retrospective cross-sectional study	CT-SMM/SMI and ASMM vs. US-RF-CSA	43 oncological surgery patients (post-surgery 65.1%), (male 72.1%, age 64.7 (SD 6.7), BMI 23.7 (SD 4.31), 67.5% digestive, TNM NR)	14/32 (CT) (11 patients without HGS)	CT-SMI vs. US-RF-CSA (cutoff 3.6 cm^2^): AUC 0.770, sensibility 70%, specificity 100%, r = 0.700 * CT-ASMM vs. US-RF-CSA (cutoff 3.29 cm^2^), AUC 0.609, sensibility 42.55%, specificity 100%, r = 0.548 *	High
López-Gómez et al., 2025 (Spain) [23]	Cross-sectional observational	US: CSA, Mi and FATi vs. CT: SMA, LMA and SM-HU	337 oncology patients on treatment (58.8% male, age 69.7 (SD 10.9), BMI 23.69 (SD 4.62), 77.4% digestive, TNM NR)	Sarcopenia 8%, low muscle mass 23.7%, dynapenia 34.7%, malnutrition 78.3%	US RF-CSA vs. CT SMA and LMA r = 0.44 and 0.47 (*p* < 0.01) US RFT vs. CT SMA & LMA r = 0.43 and 0.43 (*p* < 0.01)	Moderate

Abbreviations: ASMM: appendicular skeletal muscle mass, AUC: area under the curve, BMT: biceps muscle thickness, CSA: cross-sectional area, CT: computed tomography, FATi: ROI fat percentage, LMA: lean muscle area, Mi: ROI muscle percentage, MT: muscle thickness, NR: not reported, QMT: quadriceps muscle thickness, RF: rectus femoris, RFT: rectus femoris thickness, ROI: region of interest, SD: standard deviation, SMA: skeletal muscle area, SM-HU: skeletal muscle-Hounsfield units, SMI: skeletal muscle index, SMM: skeletal muscle mass (SMI/height), SMMI: skeletal muscle mass index, TMT: thigh muscle thickness, US: ultrasound. * Statistically significant. ^1^ Assessed using the National Institutes of Health (NIH) Quality Assessment Tool for Observational Cohort and Cross-Sectional Studies [29].

All studies were observational. There was only one prospective cohort study [13]. Another study [22] performed a secondary cross-sectional analysis of cohort studies with prospective data collection [37,38]. The remaining four studies were cross-sectional [8,23,28,31].

Four studies were developed in Spain [8,23,28,31] and two in Brazil [13,22]. All of them were recently published, in 2024 [8,28,31] or 2025 [13,22,23]. All the studies included male and female patients, but only in one of them, the female patients were more than 50% [22], with the proportion of female patients from 27.9% [28] to 53.3% [22]. Mean age was very similar among the studies, ranging from 62 [22] to 69.7 [23] years. Two of the studies included hospitalized inpatients [13,22], one of them critically ill [13], while the rest included outpatients [8,23,28,31]. Digestive tract neoplasms were the most common among the studies, including 834 patients (82.5%), and ranging from 47.7% [13] to 100% [8,31].

All studies evaluated muscle mass at L3 in computed tomography but, regarding nutritional ultrasound, one of them evaluated biceps and thigh muscle thickness (BMT, TMT) at midpoint of both [22], another study measured quadriceps and rectus femoris muscle thickness, as well as rectus femoris cross-sectional area (RF-CSA) at 2/3 of the thigh [8], a third study assessed quadriceps muscle thickness with and without compression at 1/2 and 2/3 of the thigh [13], the fourth one measured RF-CSA, circumference, and X- and Y-axis at 2/3 [31], the fifth study evaluated RF-CSA at 2/3 [28], and the last one measured RF-CSA and thickness (Y-axis) [23].

### 3.3. Risk of Bias Assessment

Following the guidance of the NIH Quality Assessment Tool for Observational Cohort and Cross-Sectional Studies [29], we found that one of the studies was evaluated as good, with low risk of bias [13], another one as poor, with high risk of bias [28], and the rest of the assessed studies as fair, with a moderate risk of bias [8,22,23,31].

The study from González-Boillos et al. [28] was considered as having high risk of bias because it only fulfilled three of the eight assessed quality criteria, and also it was from one unique center, had a small sample, and has not been peer-reviewed, among other items, which limits its methodological robustness. On the other hand, the study from de Lellis et al. [13] was considered as having low risk of bias because it fulfilled 7 of 8 quality criteria, and the remaining were not reported. The rest of the studies [8,22,23,31] were in an intermediate situation and, therefore, were considered with moderate risk of bias.

Figure 2 shows the proportion of each evaluated item across all the studies. The risk of bias analysis from every individual study is available in the Appendix A at Open Science Framework (https://osf.io/8r3vs/ (accesed on 8 November 2025)) and Appendix A.

### 3.4. Synthesis of Results

#### 3.4.1. Low Muscle Mass Prevalence and Cutoff Values

Only the study from Jiménez-Sánchez et al. [8] presented low muscle mass prevalence according to the CT and NU cutoff points [8]. They used NU DRECO [19] cutoff points of sarcopenia for hospitalized patients at nutritional risk to assess low muscle mass in outpatients with colorectal cancer. With the DRECO [19] cutoff point for RF-CSA (3.48 cm^2^ for male, 2.4 cm^2^ for female) and for RFT (9.66 mm male, 10.4 mm female), low muscle mass prevalence estimation in the Jiménez-Sánchez et al. [8] study was 13/156 (8.33%) and 16/156 (10.24%), respectively, much more closer to the estimation of muscle atrophy according to Van Vogt [39] 10/156 (6.41%) than to Dolan [40] 74/156 (48.72%) cutoff points for CT-SMI, suggesting that RF-CSA provides the best estimation with NU parameters. Data are summarized in Table 1. Jiménez-Sánchez et al. [8] also proposed an equation for estimating L3-SMA from QMT [L3-SMA (cm^2^) = −74.04 + 12.03 × QMT (cm) + 0.56 × weight (kg) + 0.7 × height (cm) + 18.86 (only in male)].

All the remaining studies showed low muscle mass prevalence according to the CT cutoff values, but with diverse cutoff references. Martin criteria [41] were used by four studies [22,23,28,31], one of them [31] together with the Prado cutoff values [42], and another one [28] did not show low muscle mass prevalence. The last study [13] used Toledo criteria [43]. Low muscle mass prevalence was 40.9%, 23.7%, 43.8%, 49.8%, and 63.6%, respectively [13,22,23,31]. In all these studies, the cutoff points for low muscle mass with NU parameters and/or correlation with CT-derived muscle mass were estimated [13,22,23,28,31].

Sousa et al. [22], in hospitalized patients with cancer, found a prevalence of low muscle mass by CT-CSA index of 49.2% and found the following cutoff points and prevalence of low muscle mass by NU parameters:BMT: 16 cm male, 12 cm female. Prevalence: 30.8%;TMT: 20 cm male, 15 cm female. Prevalence: 18.3%;TMT + BMT: 36 cm male, 43 cm female. Prevalence 48.3%. Closest to CT-CSA index.

The study from de Lellis et al. [13] estimated a cutoff point of 1.29 cm for QMT at 2/3 of the thigh with compression. Their sample of critically ill patients with cancer had a mean QMT of 1.31 (SD = 0.51), but they did not publish the prevalence of low muscle mass in the sample with their cutoff point.

López-Gómez et al. [23] proposed the cutoff points of low muscle mass for RF-CSA (2.71 cm^2^ for male, 1.88 cm^2^ for female) and for RFT (10.86 mm male, 8.67 mm female) but did not report the prevalence of in their sample of 337 outpatients with cancer.

The study from Gonzalez-Boillos et al. [28] proposed the cutoff point for low muscle mass in patients who had undergone oncological surgery for NU RF-CSA at 3.12 cm^2^ and found a mean RF-CSA of 2.96 (range 2.47–4.2), but they did not publish the prevalence in the sample either.

Guirado-Peláez et al. [31] did not propose any cutoff points. None of these studies [13,23,28,31] presented data of low muscle mass from NU parameters.

#### 3.4.2. Correlation Nutritional Ultrasound vs. Computed Tomography

Sousa et al. [22] found that NU-TMT vs. CT-CSA in male was AUC = 0.78 (r = 0.24 *) and NU-BMT + TMT AUC = 0.76 (r = 0.24 *); in female, both AUC = 0.50; confidence intervals were not reported. Both TMT and BMT + TMT independently accounted for 35% of variability in CT-CSA (R^2^ = 0.35 *). Positive predictive value (PPV) for TMT was 33.3% and 100%, in male and female patients, respectively, and negative predictive value (NPV) was 97.6% and 19.3%, respectively; for BMT + TMT, PPV was 31.2% and 78.9%, respectively, and NPV was 97.5% and 9.1%, respectively. Data are summarized in Table 1.

Jiménez-Sánchez et al. [8] reported a correlation between CT-L3-SMA and Fischer estimation [44] from QMT and theirs, respectively, of r = 0.849 * and r = 0.854 *. For the correlation between low muscle mass from the Van Vogt CT cutoff points [39] and the DRECO [19] cutoff points for RFT and RF-CSA, a slight agreement was found, k = 0.165 * and k = 0.109 (n.s.; worse results were found with the Dolan [40] CT cutoff points, k = 0.136 * and k = 0.069 (n.s.).

The study from de Lellis et al. [13] proposed a QMT cutoff point of 1.29 cm, and this showed a sensitivity of 73.2% and specificity 78.1% for diagnosing low muscle mass in critical patients with cancer, with a PPV of 85.4% (74.9–92.0) and NPV of 62.5% (31.0–72.7).

Guirado-Peláez et al. [31] found CT-SMI and CT-CSA correlated with RF-CSA, with r = 0.56 *, r = 0.63 *, respectively. González-Boillos et al. [28] reported that CT-SMI/height and CT-PMI/height correlated with RF-CSA (r = 0.700 *, r = 0.548 *, respectively).

The last study, from López-Gómez et al. [23], with the proposed RF-CSA and RFT cutoff points (see above), found an RF-CSA AUC of 0.580 and 0.417 for male and female patients, respectively, and RFT AUC of 0.593 and 0.633, respectively. RF-CSA showed high sensitivity and low specificity. RFT, on the contrary, had low sensitivity and high specificity. RF-CSA PPV was 30.87% (23.8–37.9) and NPV was 66.53% (60.1–73.0), while for RFT, PPV was 58.98% (35.1–82.9), and NPV was 78.17% (73.7–83.7). They found that RF-CSA has a moderate correlation with CT-SMA (r = 0.44 *) and CT-LMA (r = 0.47 *).

## 4. Discussion

### 4.1. Summary of Evidence

Sarcopenia should be evaluated in cancer patients as it has a great impact on cancer prognosis and treatment tolerance [6]. This evaluation must include the assessment of muscle strength and muscle mass [7].

Muscle mass is usually estimated from CT images at L3 level in cancer patients in an opportunistic way as CTs are performed only for staging and evaluating response to cancer treatment and not for evaluating body composition and nutritional status, as it is expensive and because of radiation concerns.

NU is much more available and can be repeated whenever it is necessary, as it has no adverse effects or risks for the patients. Recently, in 2022, the Global Leadership Initiative on Malnutrition (GLIM) accepted NU as a tool to assess muscle mass, with preference to anthropometry, and at the same level of other technical approaches, such as DXA, BIA, and CT, considering appropriate expertise and reference values are available [9]. GLIM considered that NU “provides potential for patient follow-up through repeated measurements” [9] but highlighted the need of “standardized techniques and protocols in terms of degree of compressibility of the skin at measurement site, and cutpoints in specific patient populations” [9].

In 2021, a Spanish group introduced NU as an emergent technique to assess body composition [45] and, in 2023, published its conceptualization, technical considerations, and standardization [46]. In the past few years (2022–2025), cutoff points of NU parameters indicating low muscle mass for hospitalized patients at risk of malnutrition (DRECO study) [19] and for oncology patients [23] have been defined, and some formulas for estimating CT-L3-SMA in non-critically ill patients by Fischer et al. (USVALID prospective study) [44] and in colorectal cancer outpatients by Jiménez-Sánchez et al. [8] have been proposed.

Once the considerations of GLIM regarding NU have been solved, it is necessary to evaluate the accuracy of the NU, comparing it with the most used technique to assess muscle mass in patients with cancer, which is CT.

Only six studies evaluating muscle mass by CT and NU in cancer patients were found. This may be because we are evaluating old technology, but with an emerging use for assessing muscle mass. The scarce number of articles and their date, all of them published last year (2024) [8,28,31] or this year (2025) [13,22,23], could support this hypothesis.

Another issue we must consider is the fact that all the studies were performed in Spain (4) [8,23,28,31] or Brazil (2) [13,22], maybe because those are the countries in which NU has been first adopted for research and for the clinical setting.

There is certain heterogeneity among the populations of the different studies. Two of them (Brazilian) included hospitalized patients with cancer [13,22], while the rest of them (Spanish) only [8,23,31] or partially [28] included outpatients. Only one of the studies included cancer patients critically ill [13]. All of them included male and female patients but in different proportions. In all the studies, digestive tract neoplasms were the most common, but, once again, with notable differences among the studies, ranging from 47.7% [13] to 100% [8,31].

The main issue regarding differences among the studies is the NU parameters evaluated. There are no differences in CT, as all the studies evaluated muscle mass at L3, but one study evaluated muscle mass by NU at the biceps and thigh [22], while the rest conducted it only at the thigh [8,13,23,28,31]. Once at the thigh, only one study evaluated muscle mass at the midpoint of the thigh [13], while all of them evaluated it at the limit between the middle and lower third of the thigh [8,13,22,23,28,31]. Only one of the studies evaluated the differences between compression and no compression of the thigh when performing NU [13]. The main parameter evaluated was RF-CSA in four studies, which also evaluated RF muscle thickness [8,23,28,31], while the remaining two studies only measured the whole quadriceps muscle thickness [13,22], showing methodological differences between Spain [8,23,28,31] and Brazil [13,22]. These differences make it difficult to compare the results and avoid us performing a meta-analysis.

Furthermore, there are differences in how to define low muscle mass with NU parameters. One of the studies, from Jiménez-Sánchez et al. [8], used the NU DRECO [19] cutoff points of sarcopenia, while others estimated the NU cutoff points of low muscle mass for patients with cancer [13,22,23,28] in different clinical scenarios, while the remaining study from Guirado-Peláez et al. [31] did not estimate or use any published NU cutoff points for low muscle mass.

Two studies reported low muscle mass prevalence according to the NU parameters. In the one from Jiménez-Sánchez et al. [8], prevalence with RF-CSA, according to the DRECO cutoff points for sarcopenia [19], was close to the prevalence estimated according to the Van Vogt [39] cutoff points for CT-SMI, while in the one from Sousa et al. [22], prevalence with NU TMT + BMT was closest to prevalence according to the CT-CSA index.

In all the studies, the assessment of low muscle mass by different NU parameters correlated with the assessment by CT, most of them significantly. Reported NU RF-CSA vs. CT AUC ranged from 0.417 (female, López-Gómez et al.) [23] to 0.78 (male, Sousa et al.) [22], and significant correlations (“r”) from 0.44 (López-Gómez et al.) [23] to 0,70 (González-Boillos et al.) [28] were also found. Sensitivity of the NU parameters for diagnosing low muscle mass in the studies that reported it ranged from 73.2% (QMT, de Lellis et al.) [13] to 78.8% (RF-CSA, López-Gómez et al.) [23], and specificity from 78.1% [13] to 30.8% [23], respectively. Both studies also reported NU parameters with a positive predictive value of 30.87% [23] to 85.4% [13] and a negative predictive value of 83.38% [23] and 62.5% [13], respectively. Sousa et al. [22] reported that NU parameters significantly accounted for 35% of variability in CT-CSA.

Risk of bias assessment has been made according to the NIH Quality Assessment Tool for Observational Cohort and Cross-Sectional Studies [29] (see Section 3.3). GRADE assessment to appraise certainty of evidence [47] was not intended to be made, and this is why it is not included in the Methods section but, according to it, as all the studies were observational, the initial level of confidence is low. The study from de Lellis et al. [13] was considered as having low risk of bias, and so, its level of confidence can be upgraded to moderate. On the contrary, the study from González-Boillos et al. [28] was considered as having high risk of bias, and this is why its level of confidence is very low. Nevertheless, also according to GRADE, NU can be strongly recommended for assessment and follow-up of low muscle mass, as “the desirable effects of intervention clearly outweigh the undesirable effects” [47].

Although it is not included in methods because it was not planned for this systematic review, a further assessment of articles was performed, according to the Synthesis Without Meta-analysis (SWiM) in systematic reviews reporting guideline [48]. Different articles were not grouped because all of them differed in populations, study design, and metric for outcomes, and no changes were made or outcomes results were transformed or synthesized or prioritized, showing them exactly as the authors reported in the articles. High heterogeneity causes among different studies have been discussed above, and SD has been included if it was reported in the original articles. For the assessment of certainty of evidence, the mentioned SD was reported when available and risk of bias assessed (see above). SWiM items table can be accessed at Open Science Framework (https://osf.io/8r3vs/ (accesed on 8 November 2025)).

In the context of the evidence, although there is much heterogeneity among the studies, NU seems a moderately reliable tool to assess muscle mass, as results can be close to that with CT, and at least a moderate correlation has been found depending on NU technique and parameters. RF-CSA and RFT at 2/3 of the thigh without compression are the most studied NU parameters to evaluate muscle mass, and reliable cutoff points for hospitalized patients at risk of malnutrition [19] and for outpatients with cancer [23] are now available. The DRECO study [19] was multi-center and evaluated 991 patients at risk of malnutrition, and the López-Gómez et al. study [23], though single-center, included 337 patients with cancer. Both studies propose cutoff points for different NU parameters and different low muscle mass categories: DRECO [19] for risk, probable, confirmed, and severe sarcopenia, and López-Gómez et al. [23] for low muscle strength, mass, and quality, and for sarcopenia. As proposed cutoff points have different clinical meanings, sensitivities, and specificities, it is necessary to choose the ones that best fit the objective: the ones with the highest sensitivity for screening purpose and those with the highest specificity for confirmation of low muscle mass. This means we should choose RF-CSA for screening and RFT for confirmation of low muscle mass, respectively, according to reported results of López-Gómez et al. [23], but further studies are needed to improve the confidence.

### 4.2. Strengths and Limitations of This Study

To our knowledge, this is the first systematic review that assesses the usefulness of NU to estimate low muscle mass in cancer patients, which is necessary to assess sarcopenia, as low muscle mass is one of its diagnostic criteria. To date, only centers with BIA can assess muscle mass whenever it is thought necessary, as CT use is opportunistic when it is performed for staging or evaluating treatment response and not for body composition assessment. NU is a widespread available technique that has only recently begun to be used with this purpose and allows repeated muscle mass assessment with more frequent scheduled timeframe or if clinical evolution make it advisable.

This systematic review was developed following the checklist from “preferred reporting items for systematic reviews and meta-analyses. 2020” (PRISMA 2020), as it is recommended. It is very updated, with the last study published as recently as on 25 September 2025 [23], and includes 1011 patients with cancer from six different studies, most of them with moderate or low risk of bias, which shows that different NU parameters are useful for assessing low muscle mass, with at least moderate concordance with CT evaluation of low muscle mass, the reference for patients with cancer.

On the contrary, among the limitations of this systematic review, there have been few studies comparing CT and NU assessment of muscle mass, all of them from 2024 or 2025, probably due to the novel use of NU with this purpose. Most of them are single-center [8,13,22,23,28], and the one from Gonzalez-Boillos et al. [28] has a low sample size, and it has the highest risk of bias, including not having been peer-reviewed. These studies are very heterogeneous in the included population (critical, hospitalized, outpatients), muscles (RF, quadriceps, biceps), area (middle, 2/3), and NU parameters evaluated (RF-CSA, RFT, TMT, BMT), making it not possible to perform a meta-analysis.

Furthermore, multi-center studies involving more countries are needed before NU can be recommended for widespread use.

Despite these limitations, in our opinion, there is enough evidence for NU to be used to assess muscle mass in the clinical setting for cancer patients, mainly in longitudinal follow-up and when CT is impractical, though an effort to standardize NU technique (location, compression), parameters, and validated cutoff points is needed. Regarding this, in light of the evidence, we recommend assessing RF-CSA and RFT at the lower 2/3 of the thigh at the first evaluation, comparing with CT assessment, and using the same NU parameters for the follow-up in a three-month scheme or whenever it is considered necessary, depending on the clinical evolution of an individual patient.

## 5. Conclusions

Muscle mass must be assessed in cancer patients, as sarcopenia is an important issue in the evolution of patients with cancer, worsening the response to cancer treatments and increasing their complications.

To date, CT is the most used technique to assess muscle mass in cancer patients but only in an opportunistic way. Nutritional ultrasound is more available, cheaper, and without risks for the patients, and it has started to be used to assess muscle mass in recent years and is being validated against other techniques to assess body composition, including CT.

In light of the evidence, NU is not as accurate as CT for assessing low muscle mass, but it seems to be a useful tool for evaluating low muscle mass, particularly, for follow-up of muscle mass through repeated measurements in cancer patients, and could be incorporated into clinical practice. In the future, NU could help to evaluate muscle quality in the clinical setting by assessing muscle steatosis and fibrosis, measuring echo intensity with the help of artificial intelligence to reduce time and variability in image analysis.

## Figures and Tables

**Figure 1 cancers-17-03683-f001:**
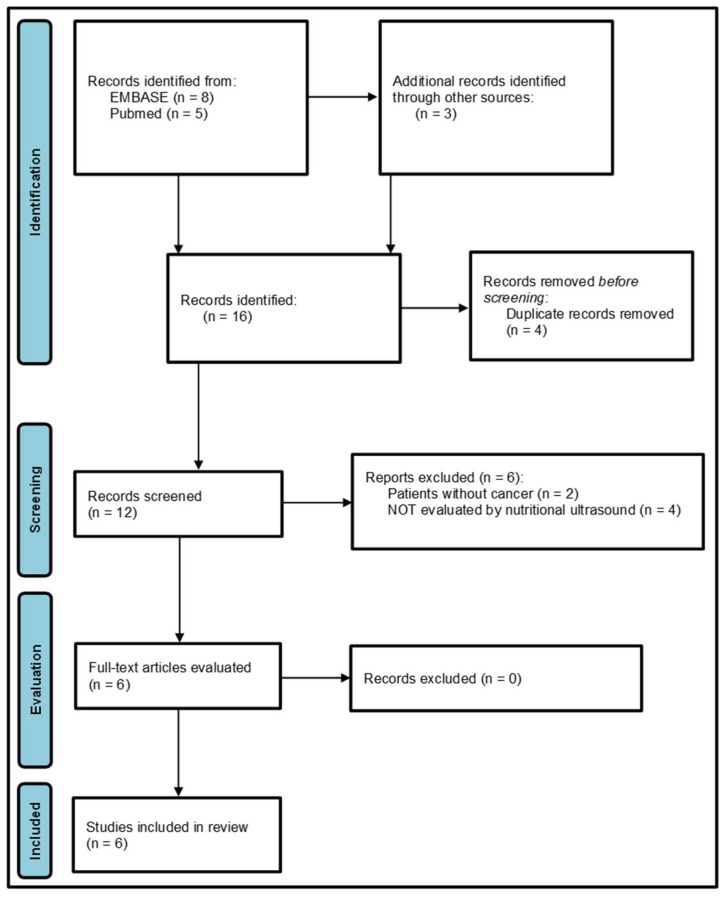
Study flow diagram following PRISMA model.

**Figure 2 cancers-17-03683-f002:**
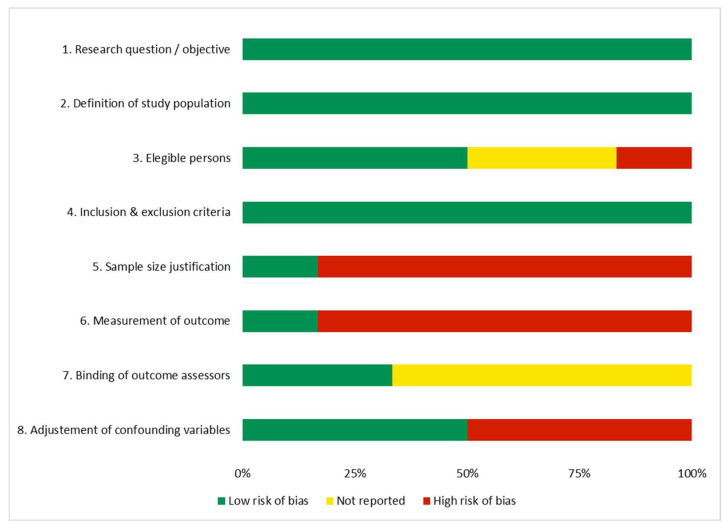
Risk of bias: judgments about each risk of bias item presented as percentages across all included studies.

## Data Availability

Data supporting reported results can be found open at https://osf.io/8r3vs/ (accesed on 8 November 2025).

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
