# Peer review of "Is Nutritional Ultrasound as Useful and Accurate as Computed Tomography to Assess Sarcopenia in Cancer Patients? A Systematic Review"

_cancers, 2025, doi:10.3390/cancers17223683_

Round 1
Reviewer 1 Report
Comments and Suggestions for Authors
This systematic review addresses an important and timely question: whether nutritional ultrasound (NU) can match computed tomography (CT) for assessing sarcopenia in cancer patients. The paper is generally well-structured and clearly written, with an OSF-registered protocol and PRISMA framing, and it synthesizes six recent studies (n=1,011) from Spain and Brazil. Nonetheless, several issues limit its interpretability and publication readiness. The objectives oscillate between accuracy, agreement, and prevalence estimation; the review would benefit from a single primary outcome (e.g., diagnostic accuracy vs. CT, expressed as AUC with 95% CIs) and clearly defined secondary outcomes (e.g., correlation coefficients, prevalence using prespecified cut-offs). At present, the synthesis commingles correlations, AUCs, sensitivity/specificity, and prevalence without a priori hierarchy or a plan for handling heterogeneity, which is substantial across populations (critical inpatients vs. outpatients), target muscles (biceps, quadriceps, rectus femoris), ultrasound protocols (site at ½ vs 2/3 of thigh, compression vs no compression), and cut-offs (DRECO, López-Gómez, study-specific thresholds). This heterogeneity justifies not performing a meta-analysis, but then the narrative synthesis should follow SWiM guidance, explicitly defining grouping, transformation of metrics, and how direction/magnitude of effects were judged; vote-counting by significance should be avoided. Consider adding a brief GRADE assessment to appraise certainty of evidence and explicitly discuss spectrum and study-level risk of bias influences on estimates.
Methods are largely compliant with PRISMA, but the information sources are inconsistent: the text first states PubMed and Scopus were searched and later reports results from EMBASE and PubMed; please reconcile databases, dates, and full search strings, and provide a PRISMA checklist. The NIH risk-of-bias assessment is informative; however, only a subset of items appears used—justify item selection or report the full tool. Extraction should specify how many reviewers resolved disagreements, whether kappa was calculated, and how missing data were handled. Where accuracy metrics are reported, add 95% confidence intervals for AUC, sensitivity, specificity, PPV/NPV, and clarify the reference standards and cut-off rationales (data-driven vs. prespecified). Because NU cut-offs differ by context (hospitalized at risk vs. oncology outpatients), the manuscript should separate analyses by clinical setting and avoid cross-applying thresholds unless justified. The conclusion that NU is “useful and accurate enough” may be overstated; a more cautious take is warranted given heterogeneity, moderate correlations (r roughly 0.44–0.70) and variable AUCs (≈0.42–0.78), with performance differing by sex and parameter.
Language is mostly clear but needs small edits for precision and consistency: use “thigh,” not “tight”; standardize “cross-sectional”; prefer “male/female” or “men/women” consistently; fix minor typographical issues (e.g., “an useful” → “a useful,” DOI commas). Ensure consistent terminology for NU parameters (RF-CSA, RFT, QMT, BMT/TMT) and clearly define each at first use. Finally, the discussion should foreground implementation: emphasize that NU is promising for longitudinal monitoring where CT is impractical, but that standardized protocols (site, compression, machine presets), externally validated cut-offs, and multicenter studies beyond two countries are prerequisites before recommending widespread clinical adoption.
Author Response
Comment 1: The objectives oscillate between accuracy, agreement, and prevalence estimation; the review would benefit from a single primary outcome (e.g., diagnostic accuracy vs. CT, expressed as AUC with 95% CIs) and clearly defined secondary outcomes (e.g., correlation coefficients, prevalence using prespecified cut-offs).
Response 1: Included in the last paragraph of Section1, Introduction.
Comment 2: At present, the synthesis commingles correlations, AUCs, sensitivity/specificity, and prevalence without a priori hierarchy or a plan for handling heterogeneity, which is substantial across populations (critical inpatients vs. outpatients), target muscles (biceps, quadriceps, rectus femoris), ultrasound protocols (site at ½ vs 2/3 of thigh, compression vs no compression), and cut-offs (DRECO, López-Gómez, study-specific thresholds). This heterogeneity justifies not performing a meta-analysis, but then the narrative synthesis should follow SWiM guidance, explicitly defining grouping, transformation of metrics, and how direction/magnitude of effects were judged; vote-counting by significance should be avoided.
Response 2: SWiM assessment done with the available items, information in page 12, section 4.1. Summary of Evidence and table can be accessed at Open Science Framework (https://osf.io/8r3vs/)
Comment 3: Consider adding a brief GRADE assessment to appraise certainty of evidence and explicitly discuss spectrum and study-level risk of bias influences on estimates.
Response 3: Included in page 11, discusión.
Comment 4: Methods are largely compliant with PRISMA, but the information sources are inconsistent: the text first states PubMed and Scopus were searched and later reports results from EMBASE and PubMed; please reconcile databases, dates, and full search strings, and provide a PRISMA checklist.
Response 4: Full search strings included in section 2.3. Search Strategy. PRISMA checklist available in supplementary material and at Open Source Framework (https://osf.io/8r3vs/).
Comment 5: The NIH risk-of-bias assessment is informative; however, only a subset of items appears used—justify item selection or report the full tool.
Response 5: Added in section 2.5. Assessment of Risk of Bias in Every Selected Study. Items included were considered the essential ones. The rest, include either items not applicable (7 timeframe, 8 level of exposure, 9 exposure measures, and 10 exposure assessed more than once, 13 loss of follow-up) as well as item 11, outcomes measures implemented consistently, which was not included because it was not discriminatory, as in every study it was done so. Examples of similar selections can be found in other peer-reviewed published systematic reviews from our group (doi: https://doi.org/10.3390/nu16121833) as well as other groups (doi: https://doi.org/10.3390/nu10111794).
Comment 6: Extraction should specify how many reviewers resolved disagreements, whether kappa was calculated, and how missing data were handled.
Response 6: Included in section 2.4. Study Selection and Data Collection.
Comment 7: Where accuracy metrics are reported, add 95% confidence intervals for AUC, sensitivity, specificity, PPV/NPV, and clarify the reference standards and cut-off rationales (data-driven vs. prespecified).
Response 7: Data asked for, when available in original articles, have been added in section 3.4.2. Correlation Nutritional Ultrasound vs. Computed Tomography. Most of confidence intervals not reported by authors. Cut-off pints for NU already included in section 3.4.1. Low muscle mass prevalence and cutoff values.
Comment 8: Because NU cut-offs differ by context (hospitalized at risk vs. oncology outpatients), the manuscript should separate analyses by clinical setting and avoid cross-applying thresholds unless justified.
Response 8: Analyses of each article have been done separately in its own clinical setting (intensive care, hospitalized at risk or outpatients with cancer) and using the same thresholds each group decided to apply.
Comment 9: The conclusion that NU is “useful and accurate enough” may be overstated; a more cautious take is warranted given heterogeneity, moderate correlations (r roughly 0.44–0.70) and variable AUCs (≈0.42–0.78), with performance differing by sex and parameter.
Response 9: Corrected in the last paragraph of section 5. Conclusions.
Comment 10: Language is mostly clear but needs small edits for precision and consistency: use “thigh,” not “tight”; standardize “cross-sectional”; prefer “male/female” or “men/women” consistently; fix minor typographical issues (e.g., “an useful” → “a useful,” DOI commas).
Response 10: Corrected. Thank you very much for your suggestions.
Comment 11: Ensure consistent terminology for NU parameters (RF-CSA, RFT, QMT, BMT/TMT) and clearly define each at first use.
Response 11: Ultrasound parameters terminology was reported exactly as it was written in the original articles to preserve authors’ intention, although we agree there is a need to standardize them and there are some of them that refer to the same reality; e.g: TMT (thigh muscle thickness) and QMT (quadriceps muscle thickness). References of articles in which terms were defined and described are provided.
Comment 12: Finally, the discussion should foreground implementation: emphasize that NU is promising for longitudinal monitoring where CT is impractical, but that standardized protocols (site, compression, machine presets), externally validated cut-offs, and multicenter studies beyond two countries are prerequisites before recommending widespread clinical adoption.
Response 12: Included in the two last paragraphs of section 4.2. Strengths and Limitations of This Study.
Reviewer 2 Report
Comments and Suggestions for Authors
Dear editor
The manuscript entitled "Is nutritional Ultrasound as Useful and Accurate as Computed Tomography to Assess Sarcopenia in Cancer Patients? A Systematic Review" evaluates effect of nutritional ultrasound (NU) on low muscle mass in patients with cancer.This manuscript can be considered for publication after major revision and addressing following comments point by point.
1- Please completely explain about search strategy in the same section. Authors explained separately about PubMed and Scopus in section 2.3 and about google scholar in 2.4
2- A section should be added to manuscript in which critically discusses about methods, selected literature,...
3- A grafical abstract which comply with subject of manuscript should be added
4- According to the PRISMA diagram, 6 studies were chosen which are too low for a review, specially discussion, interpretation and conclusion. How do you explain this challenge?
5- Author should complete conclusion and add future perspective to this section
6- Identical report should be reduced to less
than 15%
Author Response
Comment 1: Please completely explain about search strategy in the same section. Authors explained separately about PubMed and Scopus in section 2.3 and about google scholar in 2.4.
Response 1: Unified in Section 2.3. Search Strategy.
Comment 2: A section should be added to manuscript in which critically discusses about methods, selected literature,...
Response 2: Included from the 2nd paragraph in section 4.2. Strengths and Limitations of This Study.
Comment 3: A grafical abstract which comply with subject of manuscript should be added.
Response 3: The initially sent graphical abstract was corrected to fit better with subject of manuscript.
Comment 4: According to the PRISMA diagram, 6 studies were chosen which are too low for a review, specially discussion, interpretation and conclusion. How do you explain this challenge?
Response 4: Already answered in the fifth paragraph of section 4.1. Summary of Evidence: “Only 6 studies evaluating muscle mass by CT and NU in cancer patients were found. This may be because we are evaluating old technology, but with an emergent use for assessing muscle mass. The scarce number of articles and their date, all of them published last (2024) [8,28,31] or this year (2025) [13,22,23], could support this hypothesis.”
Comment 5: Author should complete conclusion and add future perspective to this section.
Response 5: Corrected and added in the last paragraph of section 5. Conclusions.
Comment 6: Identical report should be reduced to less than 15%.
Response 6: Despite the low number of studies obtained in each database (PubMed 5 and Scopus 8), 4 of them were shown as results of search strategy in both. We think it may be due to the fact that nutritional ultrasound has only recently begun to be employed in the clinical setting and there are few studies so far. Please, notice that all of them have been published last or this year.
Comment 7: The English could be improved to more clearly express the research.
Response 7: Revised and corrected.
Reviewer 3 Report
Comments and Suggestions for Authors
The article addresses the crucial clinical issue of assessing the accuracy of nutritional ultrasound compared to computed tomography in the assessment of sarcopenia in cancer patients. Given the increasing emphasis on bedside muscle assessment and the need to move beyond opportunistic CT-based measurements, the topic remains relevant. The authors synthesize current evidence, including very recent studies from 2024-2025. However, the study remains open to discussion.
- Please include full search strings (PubMed, Scopus) and specify the date range, filters, and language restrictions.
- Figure 1 (PRISMA) should be fully annotated with numbers and reasons for exclusion.
- Explain whether gray literature and ongoing clinical trials were excluded.
4- The Albumin-Myosteatosis Meter (AMG), recently proposed as a biomarker combining serum albumin concentration and CT-derived muscle radiodensity, will help contextualize the potential of NU not only as a tool for muscle quantity but also for muscle quality assessment, similar to how AMG integrates metabolic (albumin) and morphological (fat infiltration) components.
5- Consider addressing sources of heterogeneity, including muscle site selection, compression methods, and population types.
6- Since muscle quality varies with cachexia severity, please report mean BMI, tumor type distribution, and disease stage.
Comments on the Quality of English LanguageThe English could be improved to more clearly express the research.
Author Response
Comment 1: Please include full search strings (PubMed, Scopus) and specify the date range, filters, and language restrictions.
Response 1: Included in section 2.3. Search Strategy..
Comment 2: Figure 1 (PRISMA) should be fully annotated with numbers and reasons for exclusion.
Response 2: Figure 1 showing study flow diagram following PRISMA model includes reasons for exclusion as well as the number of each. We have changed “EMBASE” for “Scopus”, as it was a mistake.
Comment 3: Explain whether gray literature and ongoing clinical trials were excluded.
Response 3: In order to avoid missing other relevant studies, a secondary search strategy was developed, by screening the references included in the previously selected articles and on Google Scholar. , where an additional article was found [28]. The last article, from López-Gómez et al. [23] was found in alerts from scientific journals, when published at the end of September 2025.
Comment 4: The Albumin-Myosteatosis Meter (AMG), recently proposed as a biomarker combining serum albumin concentration and CT-derived muscle radiodensity, will help contextualize the potential of NU not only as a tool for muscle quantity but also for muscle quality assessment, similar to how AMG integrates metabolic (albumin) and morphological (fat infiltration) components.
Response 4: This review did not intend to evaluate muscle quality with NU versus CT, but only muscle mass, as it is one of the items included in GLIM criteria for diagnosing malnutrition. Assessing muscle quality with CT or NU represents a step forward after assessing muscle mass and, therefore, there is much less evidence available. Albumin-myoesteatosis measurement has only recently been developed, and articles regarding this topic are only from 2023 on. Last examples are articles published from the group of Valladolid (Spain), also authors of one of the articles found in this systematic review [23. López-Gómez J.J.; Sánchez-Lite I.; Fernández-Velasco P.; Izaola-Jauregui O.; Cebriá Á.; Pérez-López P.; González-Gutiérrez J.; Estévez-Asensio L.; Primo-Martín D.; Gómez-Hoyos E.; Jorge-Godoy E.; De Luis-Román D.A. Artificial intelligence–assisted rectus femoris ultrasound vs. L3 computed tomography for sarcopenia assessment in oncology patients: establishing diag-nostic cut-offs for muscle mass and quality. Front Nutr 2025, 1678989. https://doi.org/10.3389/fnut.2025.1678989] are: de Luis D, Primo D, Izaola O, Sánchez Lite I, López Gómez JJ. Albumin-myosteatosis gauge as a prognostic factor and its relationships with bioimpedancemetry in patients with colorectal-cancer. Clin Nutr ESPEN. 2025 Oct 9;70:281-288. doi: https://doi.org/10.1016/j.clnesp.2025.09.032 and [Albumin-myoestatosis gauge assisted by an artificial intelligence tool as a prognostic factor in patients with metastatic colorectal-cancer]. Nutr Hosp. 2025 Jun 6. Spanish. doi: https://doi.org10.20960/nh.05687
Comment 5: Consider addressing sources of heterogeneity, including muscle site selection, compression methods, and population types.
Response 5: Already addressed in paragraphs 7th to 11th in section 4.1. Summary of Evidence.
Comment 6: Since muscle quality varies with cachexia severity, please report mean BMI, tumor type distribution, and disease stage.
Response 6: Data asked for added in table 1. Summary of the studies included in the systematic review.
Comment 7: The English could be improved to more clearly express the research.
Response 7: Revised and corrected.
Round 2
Reviewer 2 Report
Comments and Suggestions for Authors
Manuscript was improved according to the comments as well and can be accepted in present form.
Reviewer 3 Report
Comments and Suggestions for Authors
I am satisfied that the authors have addressed all of my previous concerns about the article. It is now much improved and I feel that it is now suitable for publication.
Comments on the Quality of English LanguageThe English could be improved to more clearly express the research.